# Particle Cloud Generation with Message Passing Generative Adversarial Networks

**Raghav Kansal, Javier Duarte, Hao Su**
University of California San Diego
La Jolla, CA 92093, USA

**Breno Orzari, Thiago Tomei**
Universidade Estadual Paulista
São Paulo/SP - CEP 01049-010, Brazil

**Maurizio Pierini, Mary Touranakou**[*]
European Organization for Nuclear Research (CERN)
CH-1211 Geneva 23, Switzerland

**Jean-Roch Vlimant**
California Institute of Technology
Pasadena, CA 91125, USA

**Dimitrios Gunopulos**
National and Kapodistrian University of Athens
Athens 15772, Greece

## Abstract

In high energy physics (HEP), jets are collections of correlated particles produced ubiquitously in particle collisions such as those at the CERN Large Hadron Collider (LHC). Machine learning (ML)-based generative models, such as generative adversarial networks (GANs), have the potential to significantly accelerate LHC jet simulations. However, despite jets having a natural representation as a set of particles in momentum-space, a.k.a. a particle cloud, there exist no generative models applied to such a dataset. In this work, we introduce a new particle cloud dataset (JetNet), and apply to it existing point cloud GANs. Results are evaluated using (1) 1-Wasserstein distances between high- and low-level feature distributions, (2) a newly developed Fréchet ParticleNet Distance, and (3) the coverage and (4) minimum matching distance metrics. Existing GANs are found to be inadequate for physics applications, hence we develop a new message passing GAN (MPGAN), which outperforms existing point cloud GANs on virtually every metric and shows promise for use in HEP. We propose JetNet as a novel point-cloud-style dataset for the ML community to experiment with, and set MPGAN as a benchmark to improve upon for future generative models. Additionally, to facilitate research and improve accessibility and reproducibility in this area, we release the open-source JETNET Python package with interfaces for particle cloud datasets, implementations for evaluation and loss metrics, and more tools for ML in HEP development.

## 1 Introduction

Over the past decade, machine learning (ML) has become the de facto way to analyze jets, collimated high-energy sprays of particles [1] produced at the CERN Large Hadron Collider (LHC). To apply ML to jets, the most natural representation is a *particle cloud*, a variable-sized set of points in momentum space, whose radiation pattern contains rich information about the underlying physics known as quantum chromodynamics (QCD). A fundamental question is whether ML algorithms can model this underlying physics and successfully reproduce the rich high- and low-level structure in jets.

---

[*]Also at National and Kapodistrian University of Athens, Athens, Greece.

35th Conference on Neural Information Processing Systems (NeurIPS 2021).

Answering this question affirmatively has important practical applications. At the LHC, large simulated data samples of collision events[2] are generated using Monte Carlo (MC) methods in order to translate theoretical predictions into observable distributions, and ultimately perform physics analyses[3]. These samples, numbering in the billions of events, require computationally expensive modeling of the interaction of particles traversing the detector material. Recently developed generative frameworks in ML such as generative adversarial networks (GANs), if accurate enough, can be used to accelerate this simulation by potentially five orders of magnitude [2].

In this work, we advocate for a benchmark jet dataset (JetNet) and propose several physics- and computer-vision-inspired metrics with which the ML community can improve and evaluate generative models in high energy physics (HEP). To facilitate and encourage research in this area, as well as to make such research more accessible and reproducible, we release interfaces for public particle cloud datasets such as JetNet, implementations for our proposed metrics, and various tools for ML in HEP development in the JETNET library [3]. We next apply existing point cloud GANs on JetNet and find the results to be inadequate for physics applications. Finally, we develop our own message passing GAN (MPGAN), which dramatically improves results on virtually every metric, and propose it as a benchmark on JetNet.

## 2   Jets

High-energy proton-proton collisions at the LHC produce elementary particles like quarks and gluons, which cannot be isolated due to the QCD property of color confinement [4]. These particles continuously radiate or "split" into a set of particles, known as a parton shower. Eventually they cool to an energy at which they undergo the process of hadronization, where the fundamental particles combine to form more stable hadrons, such as pions and protons. The final set of collimated hadrons produced after such a process is referred to as a jet.

The task of simulating a single jet can be algorithmically defined as inputting an initial particle, which produces the jet, and outputting the final set of particles a.k.a. the jet constituents. Typically in HEP the parton shower and hadronization are steps that are simulated sequentially using MC event generators such as PYTHIA [5] or HERWIG [6]. Simulating either process exactly is not possible because of the complex underlying physics (QCD), and instead these event generators fit simplified physics-inspired stochastic models, such as the Lund string model for hadronization [7], to existing data using MC methods. The present work can be seen as an extension of this idea, using a simpler, ML-based model, also fitted to data, for generating the jet in one shot, where we are effectively trading the interpretability of the MC methods for the speed of GPU-accelerated ML generators.

**Representations.** As common for collider physics, we use a Cartesian coordinate system with the $z$ axis oriented along the beam axis, the $x$ axis on the horizontal plane, and the $y$ axis oriented upward. The $x$ and $y$ axes define the transverse plane, while the $z$ axis identifies the longitudinal direction. The azimuthal angle $\phi$ is computed with respect to the $x$ axis. The polar angle $\theta$ is used to compute the pseudorapidity $\eta = -\log(\tan(\theta/2))$. The transverse momentum ($p_\mathrm{T}$) is the projection of the particle momentum on the $(x, y)$ plane. As is customary, we transform the particle momenta from Cartesian coordinates $(p_x, p_y, p_z)$ to longitudinal-boost-invariant pseudo-angular coordinates $(p_\mathrm{T}, \eta, \phi)$, as shown in Fig. 1.

In the context of ML, jets can be represented in multiple ways. One popular representation is as images [8, 9], created by projecting each jet's particle constituents onto a discretized angular $\eta$-$\phi$ plane, and taking the intensity of each "pixel" in this grid to be a monotonically increasing function of the corresponding particle $p_\mathrm{T}$. These tend to be extremely sparse, with typically fewer than 10% of pixels nonempty [10], and the discretization process can furthemore lower the resolution.

Two more spatially efficient representations are as ordered lists or unordered sets [11, 12] of the jet constituents and their features. The difficulty with the former is that there is no particular preferred ordering of the particles—one would have to impose an arbitrary ordering such as by transverse momentum [13]. The more natural representation is the unordered set of particles in momentum space, which we refer to as a "particle cloud." This is in analogy to point cloud representations of 3D

---

[2]An event is a set of observed particles with well-defined momenta, but not generally well-defined positions.
[3]See App. A for further details on the downstream applications of simulations.

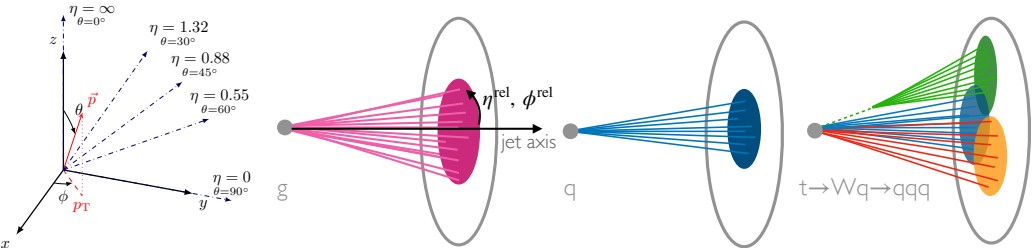

Figure 1: The collider physics coordinate system defining $(p_T, \eta, \phi)$ (left). The three jet classes in our dataset (right). Gluon (g) and light quark (q) jets have simple topologies, with q jets generally containing fewer particles. Top quark (t) jets have a complex three-pronged structure. Shown also are the relative angular coordinates $\eta^{\mathrm{rel}}$ and $\phi^{\mathrm{rel}}$, measured from the jet axis.

objects in position-space prevalent in computer vision created, for example, by sampling from 3D ShapeNet models [14].

Apart from how the samples are produced, significant differences between jets and ShapeNet-based point clouds are that, firstly, jets have physically meaningful low- and high-level features such as particle momentum, total mass of the jet, the number of sub-jets, and $n$-particle energy correlations. These physical observables are how we characterize jets, and hence are important to reproduce correctly for physics analysis applications. Secondly, unlike the conditional distributions of points given a particular ShapeNet object, which are identical and independent, particle distributions within jets are highly correlated, as the particles each originate from a single source. The independence of their constituents also means that ShapeNet-sampled point clouds can be chosen to be of a fixed cardinality, whereas this is not possible for jets, which inherently contain varying numbers of particles due to the stochastic nature of particle production.

**JetNet.** We publish JetNet [15] under the CC-BY 4.0 license, to facilitate and advance ML research in HEP, and to offer a new point-cloud-style dataset to experiment with. Derived from Ref. [16][4], it consists of simulated particle jets with transverse momenta $p_T^{\mathrm{jet}} \approx 1\,\mathrm{TeV}$, originating from gluons, light quarks, and top quarks produced in $13\,\mathrm{TeV}$ proton-proton collisions in a simplified detector. Technical details of the generation process are given in App. B. We limit the number of constituents to the 30 highest $p_T$ particles per jet, allowing for jets with potentially fewer than 30 by zero-padding. For each particle we provide the following four features: the relative angular coordinates $\eta^{\mathrm{rel}} = \eta^{\mathrm{particle}} - \eta^{\mathrm{jet}}$ and $\phi^{\mathrm{rel}} = \phi^{\mathrm{particle}} - \phi^{\mathrm{jet}} \pmod{2\pi}$, relative transverse momentum $p_T^{\mathrm{rel}} = p_T^{\mathrm{particle}}/p_T^{\mathrm{jet}}$, and a binary mask feature classifying the particle as genuine or zero-padded.

We choose three jet classes, depicted in Fig. 1, to individually target the unique and challenging properties of jets. Gluons provide a useful baseline test, as they typically radiate into a large number of particles before hadronization, largely avoiding the variable-sized cloud issue—at least with a 30 particle maximum, and have a relatively simple topology. Light quarks share the simple topology, but produce fewer final-state particles, resulting in a larger fraction of zero-padded particles in the dataset. They allow evaluation of a model's ability to handle variable-sized clouds. Finally, top quarks decay into three lighter quarks through an intermediate particle, the W boson, which each may produce their own sub-jets, leading to a complex two- or three-pronged topology—depending on whether the jet clustering algorithm captures all three or just two of these sub-jets. This results in bimodal jet feature distributions (one peak corresponding to fully merged top quark jets and the other to semi-merged, as seen in Fig. 3). Thus, top quark jets test models' ability to learn the rich global structure and clustering history of a particle cloud.

## 3   Related Work

**Generative models in HEP.** Past work in this area has exclusively used image-based representations for HEP data. One benefit of this is the ability to employ convolutional neural network (CNN) based generative models, which have been highly successful on computer vision tasks.

---

[4]This dataset was also released under the CC-BY 4.0 license.

Refs. [2, 17–20], for example, build upon CNN-based GANs, and Ref. [21] uses an auto-regressive model, to output jet- and detector-data-images.

In addition to the issues with such representations outlined in Sec. 2, the high sparsity of the images can lead to training difficulties in GANs, and the irregular geometry of the data — a single LHC detector can typically have multiple sections with differing pixel sizes and shapes — poses a challenge for CNN GANs which output uniform matrices. While these can be mitigated to an extent with techniques such as batch normalization [22] and using larger/more regular pixels [18], our approach avoids both issues by generating particle-cloud-representations of the data, as these are inherently sparse data structures and are completely flexible to the underlying geometry.

**GANs for point clouds.**    There are several published generative models in this area, however the majority exploit inductive biases specific to their respective datasets, such as ShapeNet-based [23–26] and molecular [27–29] point clouds, which are not appropriate for jets. A more detailed discussion, including some experimental results, can be found in App. C.

There do exist some more general-purpose GAN models, namely r-GAN [30], GraphCNN-GAN [31], and TreeGAN [32], and we test these on JetNet. r-GAN uses a fully-connected (FC) network, GraphCNN-GAN uses graph convolutions based on dynamic $k$-nn graphs in intermediate feature spaces, and TreeGAN iteratively up-samples the graphs with information passing from ancestor to descendant nodes. In terms of discriminators, past work has used either a FC or a PointNet [33]-style network. Ref. [34] is the first work to study point cloud discriminator design in detail and finds amongst a number of PointNet and graph convolutional models that PointNet-Mix, which uses both max- and average-pooled features, is the most performant.

We apply the three aforementioned generators and FC and PointNet-Mix discriminators as baselines to our dataset, but find jet structure is not adequately reproduced. GraphCNN's local convolutions make learning global structure difficult, and while the TreeGAN and FC generator + PointNet discriminator combinations are improvements, they are not able to learn multi-particle correlations, particularly for the complex top quark jets, nor deal with the variable-sized light quark jets to the extent necessary for physics applications.

**Message Passing Neural Networks.**    We attempt to overcome limitations of existing GANs by designing a novel generator and discriminator which can learn such correlations and handle variable-sized particle clouds. Both networks build upon the generic message-passing neural network (MPNN) [35] framework with physics-conscious design choices, and collectively we refer to them as message-passing GAN (MPGAN). We find MPGAN outperforms existing models on virtually all evaluation metrics.

### 3.1   Evaluating generative models.

Evaluating generative models is a difficult task, however there has been extensive work in this area in both the physics and computer-vision communities.

**Physics-inspired metrics.**    An accurate jet simulation algorithm should reproduce both low-level and high-level features (such as those described in Sec. 2), hence a standard method of validating generative models, which we employ, is to compare the distributions of such features between the real and generated samples[5] [2, 17–20, 36].

For application in HEP, a generative model needs to produce jets with physical features indistinguishable from real. Therefore, we propose the validation criteria that differences between real and generated sample features may not exceed those between sets of randomly chosen real samples. To verify this, we use bootstrapping to compare between random samples of only real jets as a baseline.

A practically useful set of features to validate against are the so-called "energy-flow polynomials" (EFPs) [37], which are a set of multi-particle correlation functions. Importantly, the set of all EFPs forms a linear basis for all useful jet-level features[6]. Therefore, we claim that if we observe all EFP distributions to be reproduced with high fidelity and to match the above criteria, we can conclude with strong confidence that our model is outputting accurate particle clouds.

---

[5]More details on the downstream validation procedure used in HEP analyses can be found in App. A.

[6]In the context of HEP, this means all infrared- and colinear-safe observables.

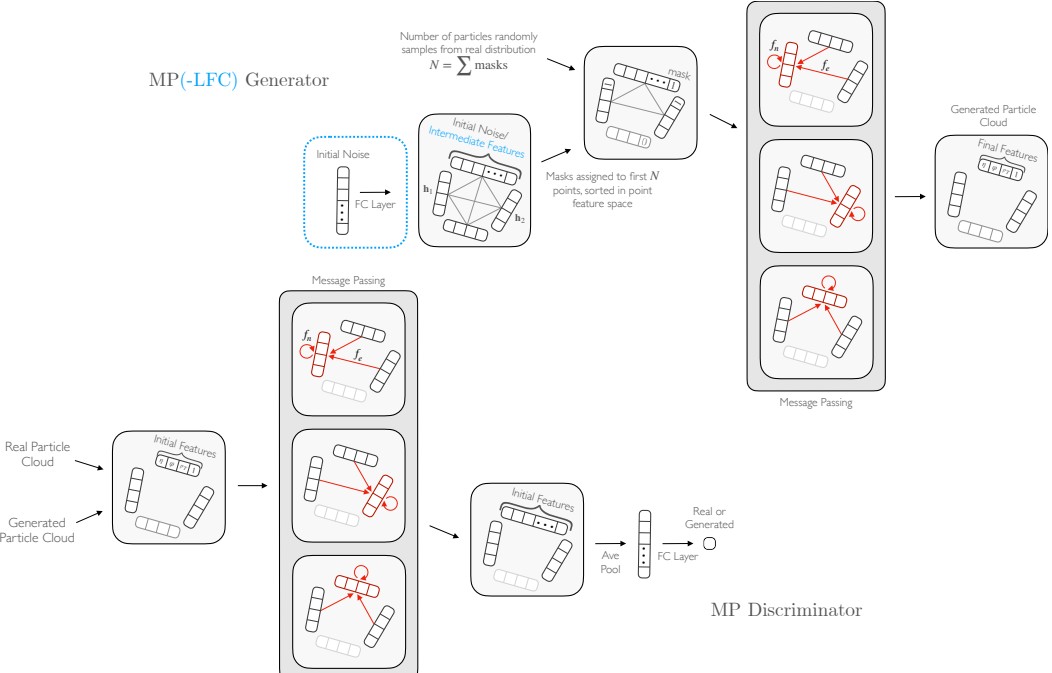

Figure 2: Top: The MP generator uses message passing to generate a particle cloud. In blue is the initial latent vector and FC layer part of the MP-LFC variant. Bottom: The MP discriminator uses message passing to classify an input particle cloud as real or generated.

**Computer-vision-inspired metrics** A popular metric for evaluating images which has shown to be sensitive to output quality and mode-collapse, though it has its limitations [38], is the Fréchet Inception Distance [39] (FID). FID is defined as the Fréchet distance between Gaussian distributions fitted to the activations of a fully-connected layer of the Inception-v3 image classifier in response to real and generated samples. We develop a particle-cloud-analogue of this metric, which we call Fréchet ParticleNet Distance (FPND), using the state-of-the-art (SOTA) ParticleNet graph convolutional jet classifier [10] in lieu of the Inception network. We note that the FPND and comparing distributions as above is conceptually equivalent, except here instead of physically meaningful and easily interpretable features, we are comparing those found to be statistically optimum for distinguishing jets.

Two common metrics for evaluating point cloud generators are coverage (COV) and minimum matching distance (MMD) [30]. Both involve finding the closest point cloud in a sample $X$ to each cloud in another sample $Y$, based on a metric such as the Chamfer distance or the earth mover's distance. Coverage is defined as the fraction of samples in $X$ which were matched to one in $Y$, measuring thus the diversity of the samples in $Y$ relative to $X$, and MMD is the average distance between matched samples, measuring the quality of samples. We use both, and due to drawbacks of the Chamfer distance pointed out in Ref. [30], for our distance metric choose only the analogue of the earth mover's distance for particle clouds a.k.a. the energy mover's distance (EMD) [40]. We discuss the effectiveness and complementarity of all four metrics in evaluating clouds in Sec. 5.

## 4 MPGAN Architecture

We describe now the architecture of our MPGAN model (Fig. 2), noting particle cloud-motivated aspects compared to its r-GAN and GraphCNN-GAN predecessors.

**Message passing.** Jets originate from a single source particle decaying and hadronizing, hence they end up with important high-level jet features and a rich global structure, known as the jet substructure [1], stemming from the input particle. Indeed any high-level feature useful for analyzing jets, such as jet mass or multi-particle correlations, is necessarily global [37]. Because of this, while past work in learning on point clouds [10, 41, 42], including GraphCNN-GAN, has used a locally

connected graph structure and convolutions for message passing, we choose a fully connected graph, equally weighting messages from all particles in the clouds. Rather than subtracting particle features for messages between particles, useful in graph convolutions to capture local differences within a neighborhood, the respective features are concatenated to preserve the global structure (the difference between particle features is also only physically meaningful if they are in the 4-vector representation of the Lorentz group). During the update step in the message passing we find it empirically beneficial to incorporate a residual connection to previous particle features.

The operation can be described as follows. For an $N$-particle cloud $J^t = \{p_1^t, \cdots, p_N^t\}$ after $t$ iterations of message passing, with $t = 0$ corresponding to the original input cloud, each particle $p_i^t$ is represented by features $\mathbf{h}_i^t$. One iteration of message passing is then defined as

$$\mathbf{m}_{ij}^{t+1} = f_e^{t+1}(\mathbf{h}_i^t \oplus \mathbf{h}_j^t), \tag{1}$$

$$\mathbf{h}_i^{t+1} = f_n^{t+1}\left(\mathbf{h}_i^t \oplus \sum_{j \in J} \mathbf{m}_{ij}^{t+1}\right), \tag{2}$$

where $\mathbf{m}_{ij}^{t+1}$ is the message vector sent from particle $j$ to particle $i$, $\mathbf{h}_i^{t+1}$ are the updated features of particle $i$, and $f_e^{t+1}$ and $f_n^{t+1}$ are arbitrary functions which, in our case, are implemented as multilayer perceptrons (MLPs) with 3 FC layers.

**Generator.**    We test two initializations of a particle cloud for the MPGAN generator: (1) directly initializing the cloud with $N$ particles with $L$ randomly sampled features, which we refer to as the MP generator, and (2) inputting a single $Z$-dimensional latent noise vector and transforming it via an FC layer into an $N \times L$-dimensional matrix, which we refer to as the MP-Latent-FC (MP-LFC) generator. The MP-LFC uses a latent space which can intuitively be understood as representing the initial source particle's features along with parameters to capture the stochasticity of the jet production process. Due to the complex nature of this process, however, we posit that this global, flattened latent space cannot capture the full phase space of individual particle features. Hence, we introduce the MP generator, which samples noise directly per particle, and find that it outperforms MP-LFC (Table 2).

**Discriminator.**    We find the MP generator, in conjunction with a PointNet discriminator, to be a significant improvement on every metric compared to FC and GraphCNN generators. However, the jet-level features are not yet reproduced to a high enough accuracy (Sec. 5). While PointNet is able to capture global structural information, it can miss the complex interparticle correlations in real particle clouds. We find we can overcome this limitation by incorporating message passing in the discriminator as well as in the generator. Concretely, our MP discriminator receives the real or generated cloud and applies MP layers to produce intermediate features for each particle, which are then aggregated via a feature-wise average-pooling operation and passed through an FC layer to output the final scalar feature. We choose 2 MP layers for both networks.

**Variable-sized clouds.**    In order to handle clouds with varying numbers of particles, as typical of jets, we introduce an additional binary "masking" particle feature classifying the particle as genuine or zero-padded. Particles in the zero-padded class are ignored entirely in the message passing and pooling operations. The MP generator adds mask features to the initial particle cloud, using an additional input of the size of the jet $N$, sampled from the real distribution, before the message passing layers based on sorting in particle feature space. Ablation studies with alternative (as well as no) masking strategies are discussed in App. E.

## 5    Experiments

**Evaluation.**    We use four techniques discussed in Sec. 3.1 for evaluating and comparing models. Distributions of physical particle and jet features are compared visually and quantitatively using the Wasserstein-1 ($W_1$) distance between them. For ease of evaluation, we report (1) the average scores of the three particle features ($W_1^{\mathrm{P}}$) $\eta^{\mathrm{rel}}$, $\phi^{\mathrm{rel}}$, and $p_{\mathrm{T}}^{\mathrm{rel}}$, (2) the jet mass ($W_1^{\mathrm{M}}$), and (3) the average of a subset of the EFPs[7] ($W_1^{\mathrm{EFP}}$), which together provide a holistic picture of the low- and high-level aspects of a jet. The $W_1$ distances are calculated for each feature between random samples

---

[7]We choose 5 EFPs corresponding to the set of loopless multigraphs with 4 vertices and 4 edges.

Table 1: $W_1$ distances between real jet mass ($W_1^{\mathrm{M}}$), averaged particle features ($W_1^{\mathrm{P}}$), and averaged jet EFPs ($W_1^{\mathrm{EFP}}$) distributions calculated as a baseline, for three classes of jets.

| Jet class | $W_1^{\mathrm{M}}$ ($\times 10^{-3}$) | $W_1^{\mathrm{P}}$ ($\times 10^{-3}$) | $W_1^{\mathrm{EFP}}$ ($\times 10^{-5}$) |
|---|---|---|---|
| Gluon | $0.7 \pm 0.2$ | $0.44 \pm 0.09$ | $0.62 \pm 0.07$ |
| Light quark | $0.5 \pm 0.1$ | $0.5 \pm 0.1$ | $0.46 \pm 0.04$ |
| Top quark | $0.51 \pm 0.07$ | $0.55 \pm 0.07$ | $1.1 \pm 0.1$ |

of 10,000 real and generated jets, and averaged over 5 batches. Baseline $W_1$ distances are calculated between two sets of randomly sampled real jets with 10,000 samples each, and are listed for each feature in Table 1. The real samples are split 70/30 for training/evaluation. We train ParticleNet for classification on our dataset to develop the FPND metric. FPND is calculated between 50,000 random real and generated samples, based on the activations of the first FC layer in our trained model[8]. Coverage and MMD are calculated between 100 real and 100 generated samples, and averaged over 10 such batches. Implementations for all metrics are provided in the JETNET package [3].

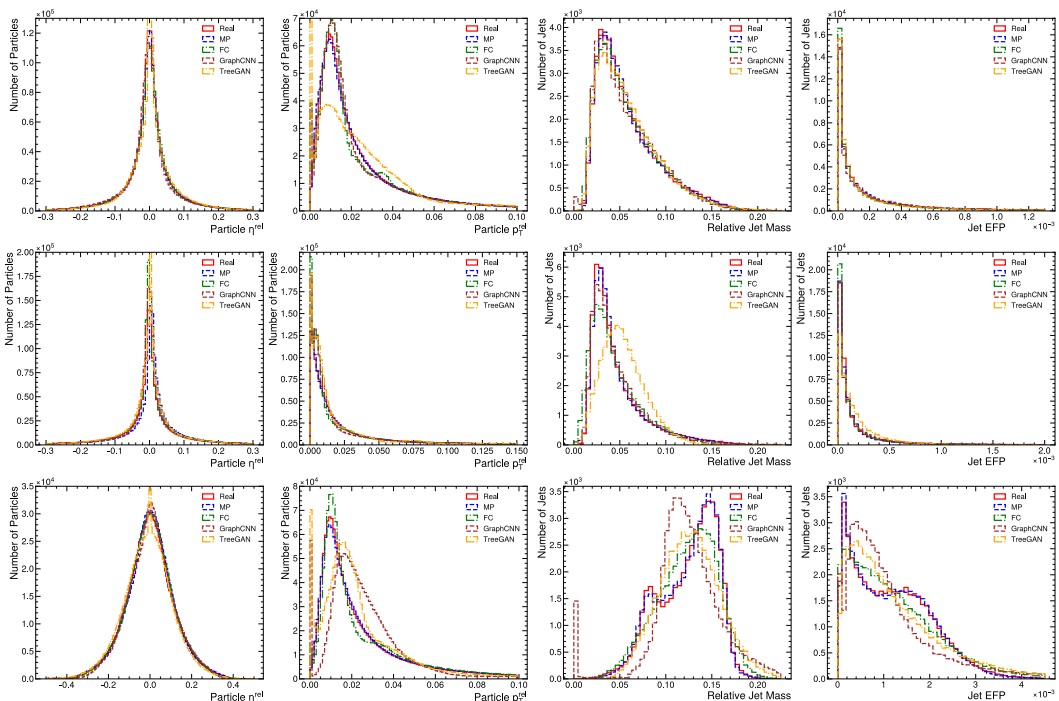

Figure 3: Comparison of real and generated distributions for a subset of jet and particle features. We use the best performing model for each of the FC, GraphCNN, TreeGAN, and MP generators, as per Table 2. Top: gluon jet features, Middle: light quark jets, Bottom: top quark jets.

**Results.** On each of JetNet's three classes, we test r-GAN's FC, GraphCNN, and TreeGAN generators with rGAN's FC and the PointNet-Mix discriminators, and compare them to MPGAN's MP generator and discriminator models, including both MP and MP-LFC generator variations. Training and implementation details for each can be found in App. D, and all code in Ref. [43].

We choose model parameters which, during training, yield the lowest $W_1^{\mathrm{M}}$ score. This is because (1) $W_1$ scores between physical features are more relevant for physics applications than the other three metrics, and (2) qualitatively we find it be a better discriminator of model quality than particle features or EFP scores. Table 2 lists the scores for each model and class, and Fig. 3 shows plots of selected feature distributions of real and generated jets, for the best performing FC, GraphCNN, TreeGAN,

---

[8]ParticleNet training details are given in App. D.3. The trained model is provided in the JETNET library [3].

Table 2: Six evaluation scores on different generator and discriminator combinations. Lower is better for all metrics except COV.

| Jet class | Generator | Discriminator | $W_1^{\mathrm{M}}$ ($\times 10^{-3}$) | $W_1^{\mathrm{P}}$ ($\times 10^{-3}$) | $W_1^{\mathrm{EFP}}$ ($\times 10^{-5}$) | FPND | COV ↑ | MMD |
|---|---|---|---|---|---|---|---|---|
| Gluon | FC | FC | $18.3 \pm 0.2$ | $9.6 \pm 0.4$ | $8.5 \pm 0.5$ | 176 | 0.24 | 0.045 |
| | GraphCNN | FC | $2.6 \pm 0.2$ | $9.6 \pm 0.3$ | $12 \pm 8$ | 61 | 0.39 | 0.046 |
| | TreeGAN | FC | $41.9 \pm 0.3$ | $69.3 \pm 0.3$ | $14.2 \pm 0.8$ | 355 | 0.19 | 0.130 |
| | FC | PointNet | $1.3 \pm 0.4$ | $1.3 \pm 0.2$ | $1.5 \pm 0.9$ | 5.0 | 0.49 | 0.039 |
| | GraphCNN | PointNet | $1.9 \pm 0.2$ | $16 \pm 6$ | $200 \pm 1000$ | 7k | 0.46 | 0.040 |
| | TreeGAN | PointNet | $1.7 \pm 0.1$ | $4.0 \pm 0.4$ | $4 \pm 1$ | 84 | 0.37 | 0.042 |
| | MP | MP | $0.7 \pm 0.2$ | $\mathbf{0.9 \pm 0.3}$ | $\mathbf{0.7 \pm 0.2}$ | **0.12** | **0.56** | 0.037 |
| | MP-LFC | MP | $\mathbf{0.69 \pm 0.07}$ | $1.8 \pm 0.2$ | $0.9 \pm 0.6$ | 0.20 | 0.54 | 0.037 |
| | FC | MP | $4.3 \pm 0.3$ | $21.1 \pm 0.2$ | $9 \pm 1$ | 368 | 0.11 | 0.085 |
| | GraphCNN | MP | $2.5 \pm 0.1$ | $9.8 \pm 0.2$ | $13 \pm 8$ | 61 | 0.38 | 0.048 |
| | TreeGAN | MP | $2.4 \pm 0.2$ | $12 \pm 7$ | $18 \pm 9$ | 69 | 0.34 | 0.048 |
| | MP | FC | $1.2 \pm 0.2$ | $3.7 \pm 0.5$ | $1.6 \pm 0.8$ | 39 | 0.44 | 0.040 |
| | MP | PointNet | $1.3 \pm 0.4$ | $1.2 \pm 0.4$ | $4 \pm 2$ | 18 | 0.53 | **0.036** |
| Light quark | FC | FC | $6.0 \pm 0.2$ | $16.3 \pm 0.9$ | $3.9 \pm 0.6$ | 395 | 0.18 | 0.053 |
| | GraphCNN | FC | $3.5 \pm 0.2$ | $15.1 \pm 0.4$ | $10 \pm 50$ | 100 | 0.25 | 0.038 |
| | TreeGAN | FC | $31.5 \pm 0.3$ | $22.3 \pm 0.4$ | $9.3 \pm 0.4$ | 176 | 0.06 | 0.055 |
| | FC | PointNet | $3.1 \pm 0.2$ | $4.5 \pm 0.4$ | $2.3 \pm 0.6$ | 17 | 0.37 | 0.028 |
| | GraphCNN | PointNet | $4 \pm 1$ | $5.2 \pm 0.5$ | $50k \pm 100k$ | 316 | 0.37 | 0.031 |
| | TreeGAN | PointNet | $10.1 \pm 0.1$ | $5.7 \pm 0.5$ | $4.1 \pm 0.3$ | 11 | 0.47 | 0.031 |
| | MP | MP | $\mathbf{0.6 \pm 0.2}$ | $4.9 \pm 0.5$ | $\mathbf{0.7 \pm 0.4}$ | 0.35 | 0.50 | 0.026 |
| | MP-LFC | MP | $0.7 \pm 0.2$ | $\mathbf{2.6 \pm 0.4}$ | $0.9 \pm 0.9$ | **0.08** | **0.52** | **0.024** |
| | FC | MP | $6.3 \pm 0.2$ | $16.5 \pm 0.2$ | $4.0 \pm 0.8$ | 212 | 0.11 | 0.070 |
| | GraphCNN | MP | $3.5 \pm 0.4$ | $15.0 \pm 0.3$ | $10 \pm 10$ | 99 | 0.26 | 0.038 |
| | TreeGAN | MP | $4.8 \pm 0.2$ | $33 \pm 6$ | $10 \pm 2$ | 148 | 0.22 | 0.041 |
| | MP | FC | $1.3 \pm 0.1$ | $4.5 \pm 0.4$ | $2.2 \pm 0.6$ | 41 | 0.37 | 0.030 |
| | MP | PointNet | $6.5 \pm 0.3$ | $23.2 \pm 0.6$ | $6 \pm 1$ | 850 | 0.18 | 0.034 |
| Top quark | FC | FC | $4.8 \pm 0.3$ | $14.5 \pm 0.6$ | $23 \pm 3$ | 160 | 0.28 | 0.103 |
| | GraphCNN | FC | $7.0 \pm 0.3$ | $8.0 \pm 0.5$ | $1k \pm 6k$ | 15 | 0.48 | 0.081 |
| | TreeGAN | FC | $17.0 \pm 0.2$ | $19.6 \pm 0.6$ | $33 \pm 2$ | 77 | 0.39 | 0.083 |
| | FC | PointNet | $2.7 \pm 0.1$ | $\mathbf{1.6 \pm 0.4}$ | $7.7 \pm 0.5$ | 3.9 | 0.56 | 0.075 |
| | GraphCNN | PointNet | $11.3 \pm 0.9$ | $30 \pm 10$ | $37 \pm 2$ | 30k | 0.39 | 0.085 |
| | TreeGAN | PointNet | $5.19 \pm 0.08$ | $9.1 \pm 0.3$ | $16 \pm 2$ | 17 | 0.53 | 0.079 |
| | MP | MP | $\mathbf{0.6 \pm 0.2}$ | $2.3 \pm 0.3$ | $\mathbf{2 \pm 1}$ | **0.37** | 0.57 | **0.071** |
| | MP-LFC | MP | $0.9 \pm 0.3$ | $2.2 \pm 0.7$ | $\mathbf{2 \pm 1}$ | 0.93 | 0.56 | 0.073 |
| | FC | MP | $6.9 \pm 0.1$ | $39.1 \pm 0.3$ | $15 \pm 1$ | 81 | 0.26 | 0.120 |
| | GraphCNN | MP | $6.7 \pm 0.1$ | $8.2 \pm 0.5$ | $40 \pm 10$ | 15 | 0.49 | 0.081 |
| | TreeGAN | MP | $13.4 \pm 0.4$ | $45 \pm 7$ | $50 \pm 30$ | 66 | 0.29 | 0.101 |
| | MP | FC | $12.9 \pm 0.3$ | $26.3 \pm 0.4$ | $46 \pm 3$ | 58 | 0.27 | 0.103 |
| | MP | PointNet | $0.76 \pm 0.08$ | $\mathbf{1.6 \pm 0.4}$ | $4 \pm 1$ | 3.7 | **0.59** | 0.072 |

and MP generators. We also provide in App. F discretized images in the angular-coordinates-plane a.k.a "jet images", however, we note that it is in general not easy to visually evaluate the quality of individual particle clouds, hence we focus on metrics and visualizations aggregated over batches of clouds. Overall we find that MPGAN is a significant improvement over the best FC, GraphCNN, and TreeGAN models, particularly for top and light quark jets. This is evident both visually and quantitatively in every metric, especially jet $W_1$s and FPND, with the exception of $W_1^{\mathrm{P}}$ where only the FC generator and PointNet discriminator (FC + PointNet) combination is more performant.

We additionally perform a latency measurement and find, using an NVIDIA A100 GPU, that MPGAN generation requires $35.7\,\mu$s per jet. In comparison, the traditional generation process for JetNet is measured on an 8-CPU machine as requiring 46ms per jet, meaning MPGAN provides a three-orders-of-magnitude speed-up. Furthermore, as noted in App. B, the generation of JetNet is significantly simpler than full simulation and reconstruction used at the LHC, which has been measured to require 12.3s [44] and 4s [45] respectively per top quark jet. Hence in practical applications we anticipate MPGAN's improvement to potentially rise to five-orders-of-magnitude.

**Real baseline comparison.**     We find that MPGAN's jet-level $W_1$ scores all fall within error of the baselines in Table 1, while those of alternative generators are several standard deviations away. This is particularly an issue with complex top quark particle clouds, where we can see in Fig. 3 none of the existing generators are able to learn the bimodal jet feature distributions, and smaller light quark clouds, where we see distortion of jet features due to difficulty reproducing the zero-padded particle features. No model is able to achieve particle-level scores close to the baseline, and only those of the FC + PointNet combination and MPGAN are of the same order of magnitude. We conclude that MPGAN reproduces the physical observable distributions to the highest degree of accuracy, but note, however, that it requires further improvement in particle feature reconstruction before it is ready for practical application in HEP.

**Architecture discussion.**     To disentangle the effectiveness of the MP generator and discriminator, we train each individually with alternative counterparts (Table 2). With the same PointNet discriminator, the GraphCNN and TreeGAN generators perform worse than the simple FC generator for every metric on all three datasets. The physics-motivated MP generator on the other hand outperforms all on the gluon and top quark datasets, and significantly so on the jet-level $W_1$ scores and the FPND. We note, however, that the MP generator is not a significant improvement over the other generators with an FC discriminator. Holding the generator fixed, the PointNet discriminator performs significantly better over the FC for all metrics. With the FC, GraphCNN, and TreeGAN generators, PointNet is also an improvement over the MP discriminator. With an MP generator, the MP discrimimator is more performant on jet-level $W_1$ and FPND scores but, on the top quark dataset, degrades $W_1^{\mathrm{P}}$ relative to PointNet.

We learn from these three things: (1) a generator or discriminator architecture is only as effective as its counterpart—even though the MPGAN combination is the best overall, when paired with a network which is not able to learn complex substructure, or which breaks the permutation symmetry, neither the generator or discriminator is performant, (2) for high-fidelity jet feature reconstruction, both networks must be able to learn complex multi-particle correlations—however, this can come at the cost of low-level feature accuracy, and (3) MPGAN's masking strategy is highly effective as both MP networks are improvements all around on light quark jets.

**Particle cloud evaluation metrics.**     We now discuss the merits of each evaluation metrics and provide suggestions for their use in future work. Fig. 4 shows correlation plots between chosen pairs of our evaluation metrics. As expected, we find W1-M and W1-EFP to be highly correlated, as they both measure learning of global jet features. For rigorous validation we suggest measuring both but for time-sensitive use-cases, such as quick evaluations during model training, W1-M should be sufficient. W1-M, FPND, and W1-P are all measuring different aspects of the generation and are relatively uncorrelated. We expect FPND overall to be the best and most discriminatory metric for evaluation, as it compares features found by a SOTA classifier to be statistically optimum for characterizing jets, while the W1 scores are valuable for their interpretability. Out of these, W1-M/W1-EFP are the most important from a physics-standpoint, as we generally characterize collisions by the high-level features of the output jets, rather than the individual particle features.

MMD and coverage are both valuable for specifically evaluating the quality and diversity of samples respectively, however we see from Fig. 4 that they saturate after a certain point, after which FPND and $W_1$ scores are necessary for stronger discrimination. We also note that in Table 2, models with low $W_1$ scores relative to the baseline have the best coverage and MMD scores as well. This indicates that the $W_1$ metrics are sensitive to both mode collapse (measured by coverage), which is expected as in terms of feature distributions mode collapse manifests as differing supports, to which the $W_1$ distance is sensitive, as well as to individual sample quality (measured by MMD), which supports our claim that recovering jet feature distributions implies accurate learning of individual cloud structure. Together this suggests that low $W_1$ scores are able validate sample quality and

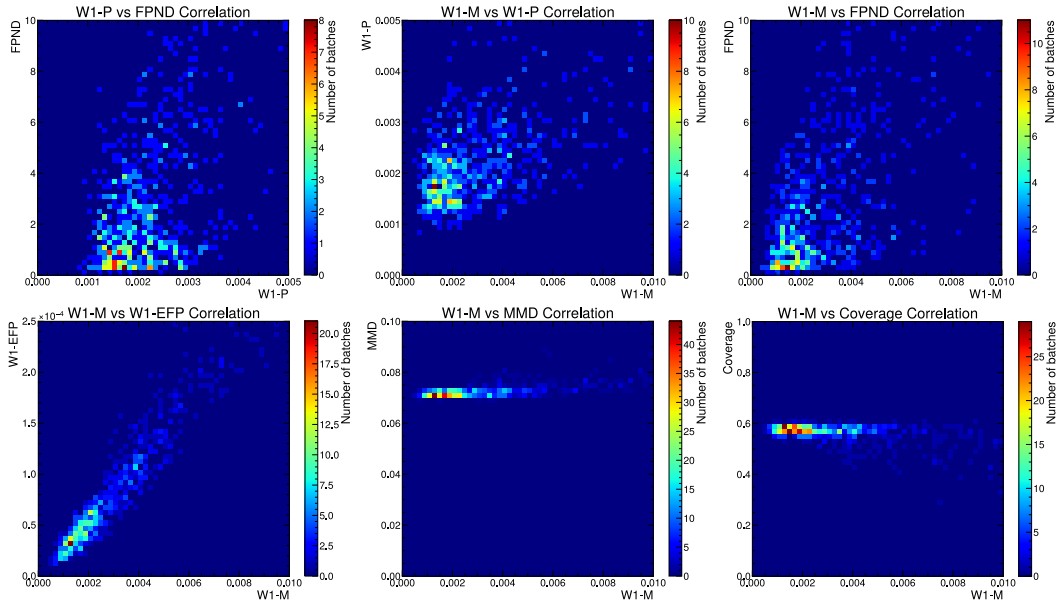

Figure 4: Correlation plots between pairs of evaluation metrics, evaluated on 400 separate batches of 50,000 MPGAN generated top quark jets.

against mode collapse, and justifies our criteria that a practical ML simulation alternative have $W_1$ scores close to the baselines in Table 2. In conclusion, for thorough validation of generated particle clouds, we recommend considering all three W-1 scores in conjunction with FPND, while MMD and coverage, being focused tests of these aspects of generation, may be useful for understanding failure modes during model development.

## 6 Summary

In this work, we publish JetNet: a novel particle cloud dataset to advance machine learning (ML) research in high energy physics (HEP), and provide a novel point-cloud-style dataset containing rich underlying physics for the ML community to experiment with. We apply existing state-of-the-art point cloud generative models to JetNet, and propose several physics- and computer-vision-inspired metrics to rigorously evaluate generated clouds. We find that existing models are not performant on a number of metrics, and fail to reproduce high-level jet features—arguably the most significant aspect for HEP. Our new message-passing generative adversarial network (MPGAN) model, designed to capture complex global structure and handle variable-sized clouds significantly improves performance in this area, as well as other metrics. We propose MPGAN as a new baseline model on JetNet and invite others to improve upon it.

**Impact**   With our JetNet dataset and library, we hope to lower the barrier to entry, improve reproducibility, and encourage development in HEP and ML. Particularly so in the area of simulation, where an accurate and fast ML particle cloud generator will have significant impact in (1) lowering the computational and energy cost of HEP research, as well as (2) increasing precision and sensitivity to new physics at the Large Hadron Collider and future colliders by providing more high-quality simulated data samples. One negative consequence of this, however, may be a loss of interpretability, and hence trustability, of the particle production generative model, which may ultimately increase uncertainties—though the metrics we propose should mitigate against this. More broadly, further advancements in the field of ML point cloud generation may result in fake visual data generation for proliferation of misinformation and impersonation/identity theft.

## Acknowledgments and Disclosure of Funding

This work was supported by the European Research Council (ERC) under the European Union's Horizon 2020 research and innovation program (Grant Agreement No. 772369). R. K. was partially supported by an IRIS-HEP fellowship through the U.S. National Science Foundation (NSF) under Cooperative Agreement OAC-1836650, and by the LHC Physics Center at Fermi National Accelerator Laboratory, managed and operated by Fermi Research Alliance, LLC under Contract No. DE-AC02-07CH11359 with the U.S. Department of Energy (DOE). J. D. is supported by the DOE, Office of Science, Office of High Energy Physics Early Career Research program under Award No. DE-SC0021187 and by the DOE, Office of Advanced Scientific Computing Research under Award No. DE-SC0021396 (FAIR4HEP). B. O and T. T are supported by grant 2018/25225-9, São Paulo Research Foundation (FAPESP). B. O was also partially supported by grants #2018/01398-1 and #2019/16401-0, São Paulo Research Foundation (FAPESP). J-R. V. is partially supported by the ERC under the European Union's Horizon 2020 research and innovation program (Grant Agreement No. 772369) and by the DOE, Office of Science, Office of High Energy Physics under Award No. DE-SC0011925, DE-SC0019227, and DE-AC02-07CH11359. D. G. is partially supported by the EU ICT-48 2020 project TAILOR (No. 952215). This work was performed using the Pacific Research Platform Nautilus HyperCluster supported by NSF awards CNS-1730158, ACI-1540112, ACI-1541349, OAC-1826967, the University of California Office of the President, and the University of California San Diego's California Institute for Telecommunications and Information Technology/Qualcomm Institute. Thanks to CENIC for the 100 Gpbs networks. Funding for cloud credits was supported by NSF Award #1904444 Internet2 supported E-CAS Exploring Clouds to Accelerate Science.

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
