# A Simulation-Based Inference in HEP

The primary motivation for producing simulations with Monte Carlo (MC) generators, and potentially by machine learning (ML) generators as well, in experimental high energy physics is to develop likelihood models for new fundamental physics theories, with unknown values of parameters of interest [46, 47]. These models are compared with experimental data, such as that collected at the LHC, to perform hypothesis tests of the theories as well as estimate and develop limits and confidence intervals for physical parameters [47, 48]. Increasing the number of simulations and their accuracy can reduce the statistical uncertainties in our models, thus allowing for higher precision measurements and potential discovery of new physics [49, 50].

In practice, in each analysis we perform rigorous checks of our simulations, including comparisons in "control regions" (selected portions of the data with a known composition) to check for MC discrepancies with real data. Mismatches are corrected via reweighting the events, i.e. unrealistic simulated jets will be given less weight in the overall analysis, and are factored into the final uncertainties in the analysis results. The same procedure should be followed for data created by generative ML methods, however, validation before this step using the metrics proposed in Sec. 3, particularly the $W_1$ scores, should mitigate the possibilities of such outliers.

# B JetNet Generation

The so-called parton-level events are first produced at leading-order using MAD-GRAPH5_aMCATNLO 2.3.1 [51] with the NNPDF 2.3LO1 parton distribution functions [52]. To focus on a relatively narrow kinematic range, the transverse momenta of the partons and undecayed gauge bosons are generated in a window with energy spread given by $\Delta p_T/p_T = 0.01$, centered at $1\,\mathrm{TeV}$. These parton-level events are then decayed and showered in PYTHIA 8.212 [5] with the Monash 2013 tune [53], including the contribution from the underlying event. For each original particle type, 200,000 events are generated. Jets are clustered using the anti-$k_T$ algorithm [54], with a distance parameter of $R = 0.8$ using the FASTJET 3.1.3 and FASTJET CONTRIB 1.027 packages [55, 56]. Even though the parton-level $p_T$ distribution is narrow, the jet $p_T$ spectrum is significantly broadened by kinematic recoil from the parton shower and energy migration in and out of the jet cone. We apply a restriction on the measured jet $p_T$ to remove extreme events outside of a window of $0.8\,\mathrm{TeV} < p_T < 1.6\,\mathrm{TeV}$ for the $p_T = 1\,\mathrm{TeV}$ bin. This generation is a significantly simplified version of the official simulation and reconstruction steps used for real detectors at the LHC, so as to remain experiment-independent and allow public access to the dataset.

# C Point Cloud Generative Models

Apart from the GAN models discussed in Sec. 3, there are several published generative models for point clouds which we argue are not applicable to jets.

## C.1 ShapeNet Point Clouds

A number of successful generative models exploit a key inductive bias of ShapeNet-based clouds: that the individual distributions of sampled points conditioned on a particular object are identical and independent (the i.i.d assumption). This assumption allows for hierarchical generative frameworks, such as Point-Cloud-GAN (PCGAN) [23], which uses two networks: one to generate a latent object-level representation, and a second to sample independent points given such a representation. The PointFlow [24] and Discrete PointFlow [25] models use a similar idea of sampling independently points conditioned on a learnt latent representation of the shape, but with a variational autoencoder (VAE) framework and using normalizing flows for transforming the sampled points.

This hierarchical-sampling approach is appealing for ShapeNet clouds, however, as discussed in Sec. 2 the key i.i.d. assumption is not applicable to jets with their highly correlated particle constituents. In fact, in contrast to ShapeNet objects which have a structure independent of the particular sampled cloud, jets are entirely defined by the distribution of their constituents.

Another model, ShapeGF [26], uses an approach of again sampling points independently from a prior distribution, but transforming them to areas of high density via gradient ascent, maximizing a

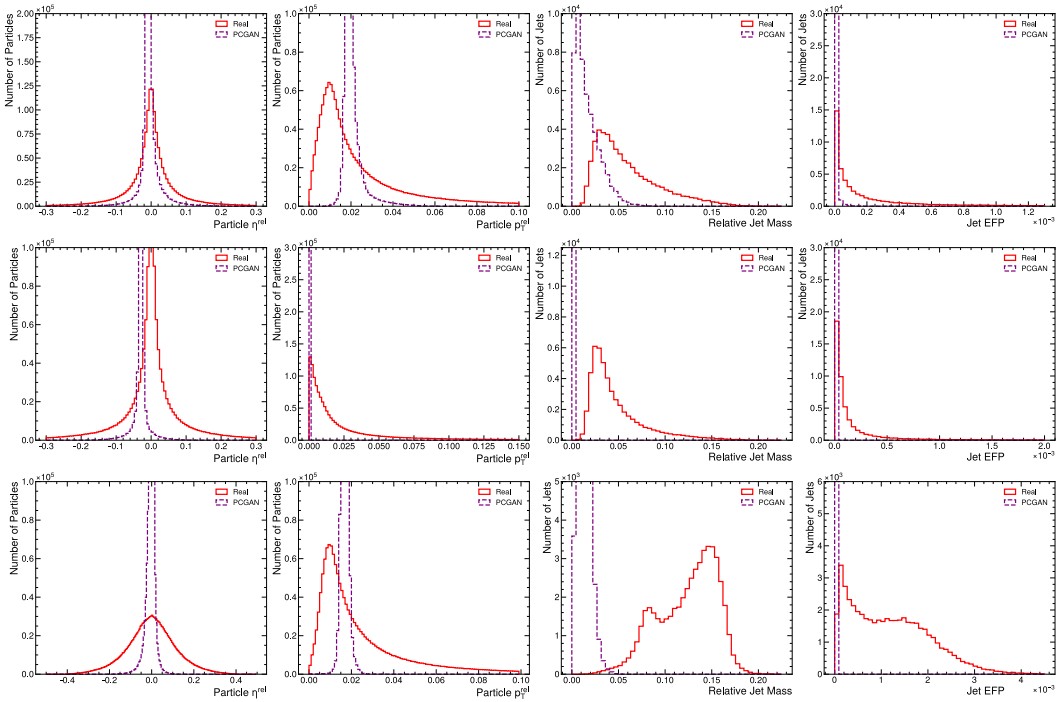

Figure 5: Comparison of real and PCGAN-generated distributions for a subset of jet and particle features. Top: gluon jet features, Middle: light quark jets, Bottom: top quark jets.

learnt log-density concentrated on an object's surface. This approach suffers as well from the i.i.d. assumption in the context of jets, and additionally, unlike for ShapeNet point clouds, there is no such high-density region in momentum-space where particles tend to be concentrated, so learning and maximizing a log-density is not straightforward.

To support our overall claim of the inviability of the i.i.d. assumption for particle clouds, we train a PCGAN model on JetNet and show the produced feature distributions in Fig. 5. We can see, as expected, while this network is partially reproducing the particle feature distributions, it is entirely unable to learn the jet-level structure in particle clouds.

## C.2 Molecular Point Clouds

3D molecules are another common point-cloud-style data structure, and there have been developments in generative models in this area as well. Kohler et al. [27] introduce physics-motivated normalizing flows equivariant to rotations around the center of mass, i.e. the SO(N) symmetries, for generating point clouds. This is appealing as normalizing flows give access to the explicit likelihood of generated samples, and having an architecture equivariant to physical symmetries such as 3D rotations can improve the generalizability and interpretability of the model. Since jets are relativistic, however, we require an architecture equivariant to the non-compact SO(3, 1) Lorentz group, to which this model has not been generalized yet. Simm et al. [28] present a reinforcement-learning-based approach for generating 3D molecules, using an agent to iteratively add atoms to a molecule and defining the reward function as the energy difference between the new molecule and the old with the new atom at the origin. This reward function is not directly applicable to jets. where particle distributions are based on the QCD dynamics rather than on minimizing the total energy. Finally, Gebauer et al. [29] introduce G-SchNet, an autoregressive model for producing molecules represented as point clouds, iteratively adding one atom at a time based on the existing molecule. Their iterative procedure however was proposed for point clouds of at most nine atoms, and does not scale well in terms of time to larger clouds.

Overall, all the models discussed heavily incorporate inductive biases which are specific to their respective datasets and don't apply to JetNet. However, they are extremely interesting approaches

nonetheless, and adapting them with jet-motivated biases should certainly be explored in future work. Indeed, a significant contribution of our work is publishing a dataset which can facilitate and hopes to motivate such development.

## D  Training and Implementation Details

PyTorch code and trained parameters for models in Table 2 are provided in the MPGAN repository [43]. Models were trained and hyperparameters optimized on clusters of NVIDIA GeForce RTX 2080 Ti, Tesla V100, and A100 GPUs.

### D.1  MPGAN

We use the least squares loss function [57] and the RMSProp optimizer with a two time-scale update rule [39] with a learning rate (LR) for the discriminator three times greater than that of the generator. The absolute rate differed per jet type. We use LeakyReLU activations (with negative slope coefficient 0.2) after all MLP layers except for the final generator and discriminator outputs where tanh and sigmoid activations respectively are applied. We attempted discriminator regularization to alleviate mode collapse via dropout [58], batch normalization [22], a gradient penalty [59], spectral normalization [60], adaptive competitive gradient descent [61] and data augmentation of real and generated graphs before the discriminator [62–64]. Apart from dropout (with fraction 0.5), none of these demonstrated significant improvement with respect to mode dropping or cloud quality.

We use a generator LR of $10^{-3}$ and train for 2000 epochs for gluon jets, $2 \times 10^{-3}$ and 2000 epochs for top quark jets, and $0.5 \times 10^{-3}$ and 2500 epochs for light quark jets. We use a batch size of 256 for all jets.

### D.2  rGAN, GraphCNNGAN, TreeGAN, and PointNet-Mix

For rGAN and GraphCNNGAN we train two variants: (1) using the original architecture hyperparameters in Refs. [30, 31] for the 2048-node point clouds, and (2) using hyperparameters scaled down to 30-node clouds—specifically: a 32 dimensional latent space, followed by layers of 64, 128, and 90 nodes for r-GAN, or followed by two graph convolutional layers with node features sizes of 32 and 24 respectively for GraphCNN-GAN. The scaled-down variant performed better for both models, and its scores are the ones reported in Table 2. For TreeGAN, starting from single vertex—in analogy with a jet originating from a single particle—we use five layers of up-sampling and ancestor-descendant message passing, with a scale-factor of two in each and node features per layer of 96, 64, 64, 64, and 64 respectively. LRs, batch sizes, loss functions, gradient penalties, optimizers, ratios of critic to generator updates, activations, and number of epochs are the same as in the original paper and code. We use the architecture defined in [34] for the PointNet-Mix discriminator.

### D.3  FPND

Apart from the number of input particle features (three in our case, excluding the mask feature), we use the original ParticleNet architecture in Ref. [10]. We find training with the Adam optimizer, LR $10^{-4}$, for 30 epochs outperformed the original recommendations on our dataset. Activations after the first fully connected layer, pre-ReLU, are used for the FPND measurement.

### D.4  PCGAN

We use the original PCGAN implementation for the sampling networks and training, with a 256-dimensional latent object representation. For the latent code GAN we use a 3 layer fully connected network for both the generator, with an input size of 128 and intermediate layer sizes of 256 and 512, and discriminator, with intermediate layer sizes of 512 and 256, trained using the Wasserstein-GAN [65] loss with a gradient penalty.

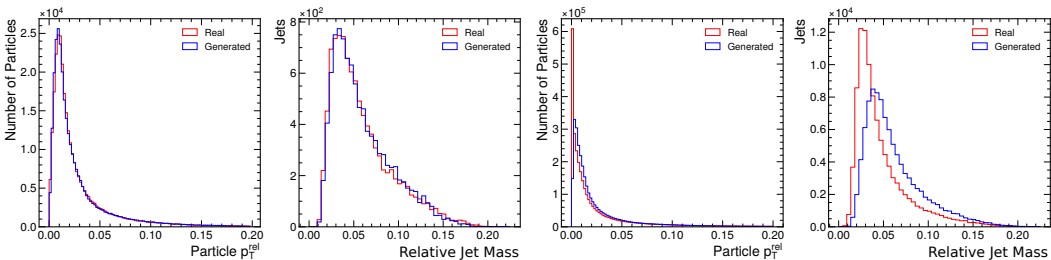

Figure 6: Particle $p_T^{\text{rel}}$ and relative jet mass distributions of real jets and those generated by MPGAN without our masking strategy. Left: gluon, right: light quark jets. We see that while for gluon jets the generator learns distributions correctly, it struggles to learn the discontinuous spike, due to the zero-padded particles, in the light quark $p_T^{\text{rel}}$ distribution. This also leads to a distorted mass distribution.

## E    Masking Strategies

In JetNet, jets with fewer than 30 particles are zero-padded to fill the 30-particle point cloud. Such zero-padded particles pose a problem for the generator, which is not able to learn this sharp discontinuity in the jet constituents (Fig. 6).

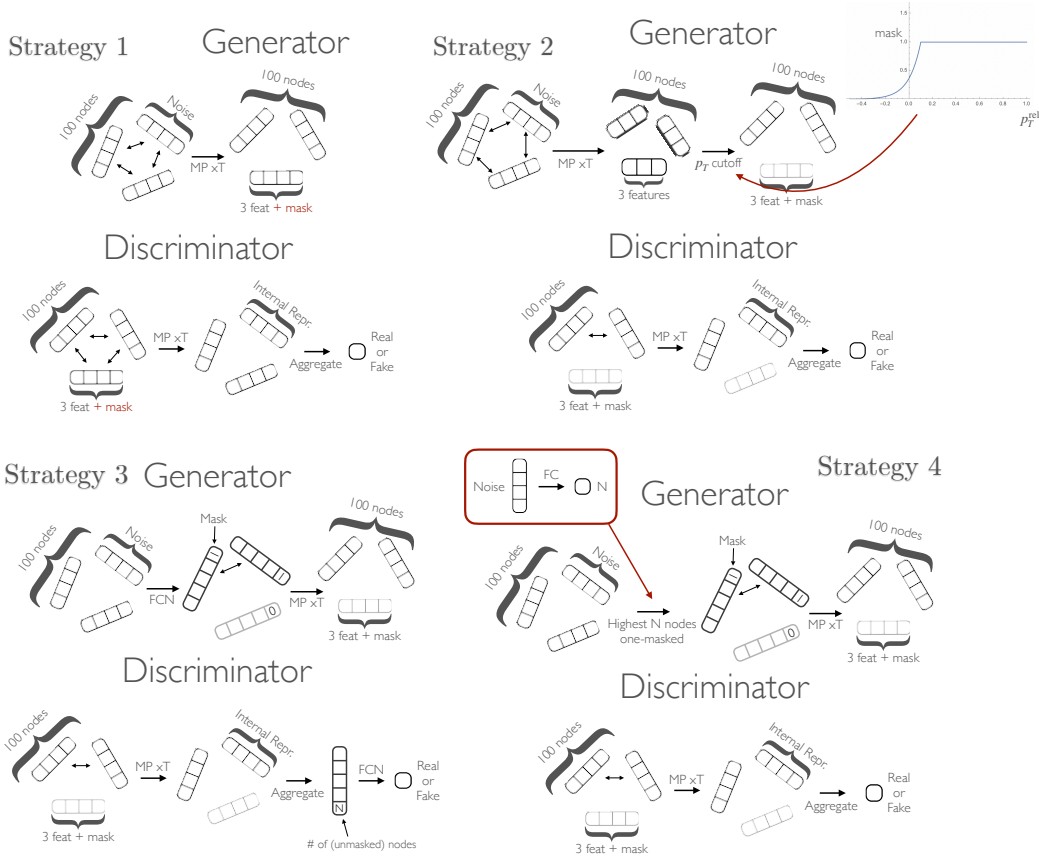

Figure 7: The four alternative masking strategies which we test.

To counter this issue, we experiment with five masking strategies, out of which the one described in Sec. 4 was most successful. The four alternatives, which all involve the generator learning the mask without any external input, are shown in Fig. 7.

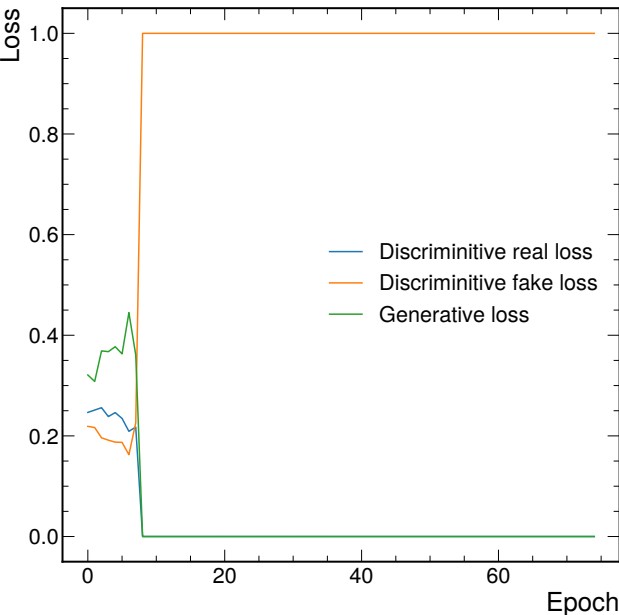

Figure 8: Loss curve of a training on light quark jets with masking strategy 3, typical of loss curves with all four strategies.

Strategy 1 treats the mask homogeneously as an extra feature to learn. A variation of this weights the nodes in the discriminator the mask. In strategy 2, a mask is calculated for each generated particle as a function of its $p_T^{rel}$, based on an empirical minimum cutoff in the dataset. In particular, both a Heaviside-step-function and a continuous mask function as in the figure are tested. The standard MP discriminator, as described in Sec. 4, is used. Strategy 3 sees the generator applying an FC layer per particle in the initial cloud to learn their respective masks, with both the MP discriminator, as well as a variant with the number of unmasked nodes in the clouds added as an extra feature to the FC layer. In strategies 1 and 3 we test both binary and continuous masks. Finally, in strategy 4, we train an auxiliary network to choose a number of particles to mask (as opposed to sampling from the real distribution), which is then passed into the standard MP generator.

We find that all such strategies are unable to produce accurate light quark jets, and in fact trainings for each diverge in the fashion depicted in Fig. 8, even using each discriminator regularization method mentioned in App. D). We conclude that learning the number of particles to produce is a significant challenge for a generator network, but is a relatively simple feature with which to discriminate between real and fake jets. To equalize this we use the strategy in Sec. 4 where the number of particles to produce is sampled directly from the real distribution, removing the burden of learning this distribution from the generator network.

## F   Jet Images

Figs. 9–11 show samples of real and generated "jet images": discretized representations of jets in the angular-coordinate-plane.

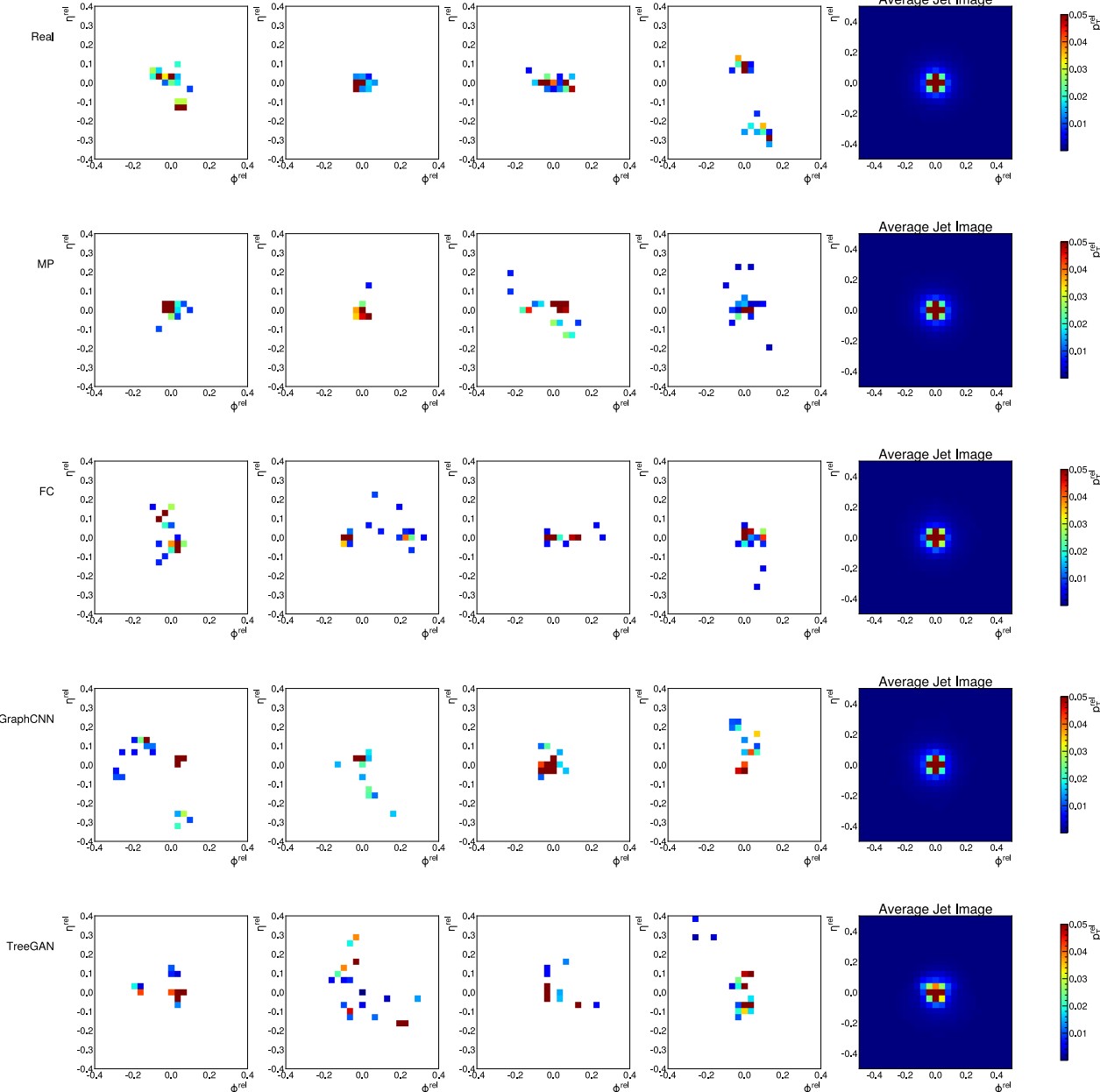

Figure 9: Random samples of discretized images in the $\eta^{\mathrm{rel}} - \phi^{\mathrm{rel}}$ plane, with pixel intensities equal to particle $p_{\mathrm{T}}^{\mathrm{rel}}$, of real and generated gluon jets (left), and an average over 10,000 such sample images (right).

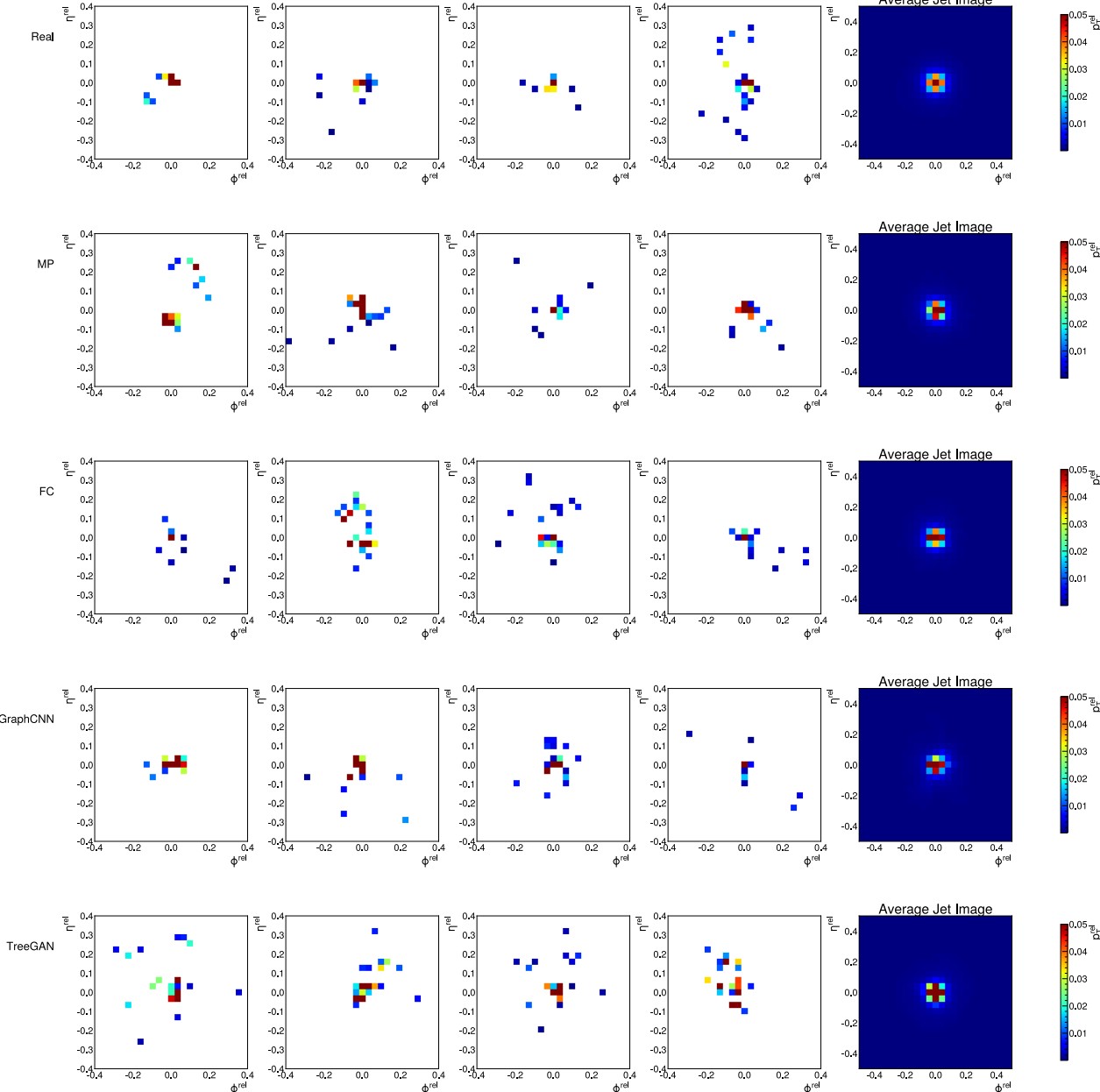

Figure 10: Random samples of discretized images in the $\eta^{\mathrm{rel}} - \phi^{\mathrm{rel}}$ plane, with pixel intensities equal to particle $p_{\mathrm{T}}^{\mathrm{rel}}$, of real and generated light quark jets (left), and an average over 10,000 such sample images (right).

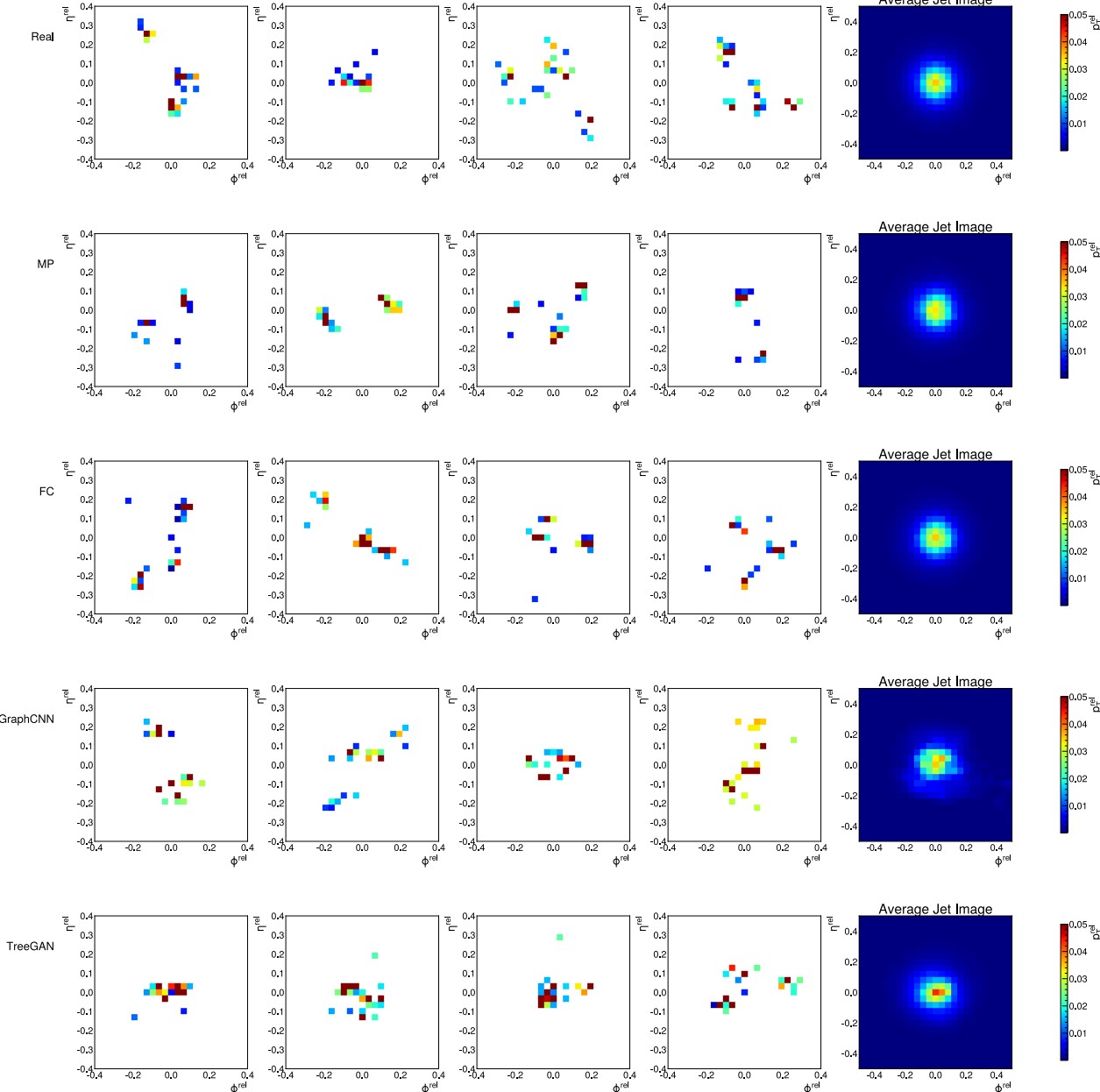

Figure 11: Random samples of discretized images in the $\eta^{\mathrm{rel}} - \phi^{\mathrm{rel}}$ plane, with pixel intensities equal to particle $p_T^{\mathrm{rel}}$, of real and generated top quark jets (left), and an average over 10,000 such sample images (right).