# OpenReview forum: "Particle Cloud Generation with Message Passing Generative Adversarial Networks"
_NeurIPS.cc/2021/Conference — NeurIPS 2021 Poster_

### Official Review · Reviewer_He4i · 2021-07-12

**Rating:** 7
**Confidence:** 4

**Summary:**

The paper introduces the new particle cloud dataset JetNet and corresponding evaluation metrics for ML in high energy physics. Several existing point-cloud generative models are evaluated and a new baseline model for the dataset is introduced.

**Limitations And Societal Impact:**

Yes

**Main Review:**

The introduction of JetNet is well-motivated and tackles a challenging problem in high energy physics.  The proposal of a message passing GAN (MPGAN) designed for this task shows significant improvement upon existing GAN approaches for the generation of point sets. As the authors note, MPGAN is still not sufficient for real-world applications such that JetNet presents a challenging task that might need novel approaches to achieve sufficient precision with a generative model.

The paper is clearly written and structured in a way that nicely introduces the problem setting. The discussion of design choices and merits/limitations of the metrics as well as the evaluation of all the models on three different tasks (gluon, light quarks, and top quarks) is thorough.

Minor issues:
- The comparison focuses very much on GANs. There is also some related work on generating 3d molecules based on autoregressive models and normalizing flows, which deals with similar issues, e.g. variable number of atoms and rotational symmetries (Gebauer et al., NeurIPS 2019, Köhler et al., ICML 2020, Simm et al., ICLR 2021). A discussion on whether these kinds of models might be applicable would be interesting.
- The fonts in the figures is quite small and hard to read (in particular the labels in Fig 2 and labels in Fig. 4.
- It would be nice for the order of gluon, light quarks and top quarks to be consistent across plots and tables.

**Time Spent Reviewing:**

3

---

> ### Author Response · Authors · 2021-08-11
> **Response to Reviewer 3**
>
> We greatly appreciate the positive feedback as well the helpful suggestion regarding discussing models for 3D molecules. We will incorporate the paragraph below in our related work section.
>
> “Kohler et al. introduce physics-motivated normalizing flows equivariant to rotations around the center of mass [1], i.e. the SO(N) symmetries, for generating point clouds. This is appealing as normalizing flows give access to the explicit likelihood of generated samples, and having an architecture equivariant to physical symmetries such as 3D rotations can improve the generalizability and interpretability of the model. Since jets are relativistic, we require an architecture equivariant to the non-compact SO(3, 1) Lorentz group, to which this model has not been generalized yet. Simm et al. [2] present a reinforcement-learning-based approach for generating 3D molecules, using an agent to iteratively add atoms to a molecule and defining the reward function as the energy difference between the new molecule and the old molecule with the new atom at the origin. This reward function is not directly applicable to jets where particle distributions are based on the QCD dynamics rather than minimizing the total energy. Both generalizing the equivariance from SO(3) to SO(3, 1), and developing a suitable reward function for jet generation are interesting areas for future development, which we hope to facilitate by publishing JetNet. Finally, Gebauer et al. [3] introduce G-SchNet, an autoregressive model for producing molecules represented as point clouds, iteratively adding one atom at a time based on the existing molecule. Their iterative procedure however was proposed for point clouds of at most 9 atoms, and does not scale well in terms of time to larger clouds.
>
> [1] Kohler et al. Equivariant Flows: Exact Likelihood Generative Learning for Symmetric Densities, ICML 2020
>
> [2] Simm et al. Symmetry-Aware Actor-Critic For 3D Molecular Design. ICLR 2021
>
> [3] Gebauer et al. Symmetry-adapted generation of 3d point sets for the targeted discovery of molecules, NeurIPS 2019”
>
> &nbsp;&nbsp;
>
> As noted above, one of the suggested autoregressive models, G-SchNet, has an iterative procedure (originally proposed for point clouds of at most 9 atoms), and does not scale well in terms of time to larger clouds. We will additionally add a timing study of our adaptation of G-SchNet to JetNet to the supplementary materials to support this.
>
> Formatting suggestions will also be incorporated into the final manuscript.

---

### Official Review · Reviewer_Q2fq · 2021-07-17

**Rating:** 6
**Confidence:** 4

**Summary:**

This paper proposes to explore an interesting application of existing point cloud generative models to the problem of generation of jets: high energy sprays of particles from elementary particle colliders. Jets are analogous in a way to point clouds with a difference in considered per-object features: per-particle momenta are considered for jets contrary to per-point 3D coordinates for point clouds. With sufficient level of quality such generative models could potentially be used instead of costly Monte Carlo simulations.

The authors explore existing point cloud gan-based generative models and propose an adaptation of existing message-passing networks to be used in generators and discriminators of resulting particle cloud gans. Several metrics are proposed for evaluation, including adaptation of existing point cloud generation task metrics, and the distance between momenta distributions for real and generated jets.

**Limitations And Societal Impact:**

The impact and limitations were briefly discussed.

**Main Review:**

Pros:
* The proposed application of point cloud generative models is novel and interesting. The authors promise to release the dataset to encourage researchers to contribute to the task.
* Both qualitative and quantitative evaluations show that the proposed message passing generators and discriminators yield better performance compared to straightforward application of point cloud specific alternatives.

Cons:
* The novelty of the approach is somewhat limited, since the overall proposed model is a combination of existing point cloud generative approaches and prior message passing networks. However, given the novel application context I do not think it is a significant drawback.
* Related work section could be improved by incorporation of existing point cloud generative models based on probabilistic modeling [1, 2, 3]. These approaches naturally overcome the issues of models with fixed point cloud size outputs and, in general, might be of interest to the authors.

The overall quality of the submission is sufficient and even though the novelty is limited, I think it is an interesting enough application context to allow it to be published to draw attention to the task.

Questions:
1) What was the number of message passing steps used in the considered architectures?
2) Were there any ablations done for the configuration of the final generators and discriminators in the proposed MPGAN?
3) You mention that one of the problems with application of ML-based generative models to physics in the lack of interpretability. You try to justify the use of such approaches if they would achieve the quality levels comparable to real baselines. Does these baselines give any guarantees on the objectivity of the resulting jet data? In case of GANs there are no such guaranties. Even if the mean metric aggregated over the whole evaluation dataset is relatively close to the aggregated metric calculated on the baseline data, there could be generated samples, which could be completely unrealistic (models are not strictly constrained to exclude them). If applied in the real-life scenarios, how generative approaches can overcome this issue?


[1] Yang, G., Huang, X., Hao, Z., Liu, M.Y., Belongie, S., Hariharan, B.: PointFlow: 3D point cloud generation with continuous normalizing flows. In ICCV’19.

[2] Klokov, R., Boyer, E., Verbeek, J.: Discrete Point Flow Networks for Efficient Point Cloud Generation. In ECCV’20.

[3] Cai, R., Yang, G., Averbuch-Elor, H., Hao, Z., Belongie, S., Snavely, N., Hariharan, B.: Learning Gradient Fields for Shape Generation. In ECCV’20.

**Time Spent Reviewing:**

5

---

> ### Author Response · Authors · 2021-08-11
> **Response to Reviewer 2**
>
> Thank you for the positive feedback regarding the novel application, as well as the thoughtful critiques. We address the questions and concerns raised individually below.
>
> &nbsp;&nbsp;
>
> **Novelty of approach**
>
> We agree that the high energy physics application context is one of the most significant contributions of this work. However, we note that the novelty in our approach also lies in the new application-informed design choices and inductive biases which result in a significant improvement over a naive application of existing architectures. Another contribution is the masking strategy which allows variable-sized generation while still retaining a graph-based architecture instead of needing to assume independence between particles, such as in the PC-GAN [1] and PointFlow [2] approaches. We will clarify this in the Architecture section.
>
> [1] Li et al., Point Cloud GAN, ICLR 2019 Workshop.
>
> [2] Yang, G., Huang, X., Hao, Z., Liu, M.Y., Belongie, S., Hariharan, B.: PointFlow: 3D point cloud generation with continuous normalizing flows. In ICCV’19.
>
> &nbsp;&nbsp;
>
> **Related work**
>
> Based on the suggested references, we will add the following paragraph below to Sec. 2. Despite the limitations mentioned below, we will add experimental results (currently in progress) using Discrete PointFlow and ShapeGF in a new Appendix section.
>
> “The PointFlow and Discrete PointFlow [3] models use a variational autoencoder framework for the generation of point clouds, where points are sampled independently, conditioned on a latent representation of the shape, and transformed using continuous normalizing flows and affine coupling layers respectively. These approaches are appealing as they give access to the explicit log-likelihood, and as they allow for variable-sized clouds. However, the idea of hierarchical sampling and the key i.i.d. assumption for the points is not applicable to particle clouds. Individual particles are not distributed independently in a particular jet but are highly correlated; in fact, in contrast to ShapeNet objects which have a structure independent of the particular sampled cloud (thus allowing for hierarchical sampling), jets are entirely defined by the distribution of their particle constituents. The ShapeGF [4] model similarly samples points independently from a prior distribution but transforms them to areas of high density via gradient ascent, maximizing a learnt log-density concentrated on an object’s surface. This approach suffers as well from the i.i.d. assumption in the context of jets, and additionally, unlike for ShapeNet point clouds, there is no such high-density region in momentum-space where particles tend to be concentrated, so learning and maximizing a log-density is not straightforward. Overall, all three models heavily incorporate inductive biases which are specific to the ShapeNet dataset, and don’t apply to JetNet, but are extremely interesting approaches nonetheless, and adapting them with jet-motivated inductive biases should certainly be explored in future work. Indeed, a significant contribution of our work is publishing a dataset which can facilitate and hopes to motivate such development.
>
> [3] Klokov, R., Boyer, E., Verbeek, J.: Discrete Point Flow Networks for Efficient Point Cloud Generation. In ECCV’20.
>
> [4] Cai, R., Yang, G., Averbuch-Elor, H., Hao, Z., Belongie, S., Snavely, N., Hariharan, B.: Learning Gradient Fields for Shape Generation. In ECCV’20.”
>
> &nbsp;&nbsp;
>
> **Questions**
>
> 1. Number of message passing steps
>
> We note in Section 4 that we use 2 message passing layers for the generator and discriminator each in MPGAN, and in Appendix A that for GraphCNN-GAN we test both the original published architecture and one scaled down to our 30-particle clouds. Specifically for the latter the original architecture used 5 graph convolutional layers and our scaled-down used 2 with output sizes of 24 and 3 respectively. These details will be added to Appendix A, and additionally our code for all, including existing, models will be made public for clarity and reproducibility.
>
>
> 2. Ablation studies
>
> Yes, we performed several ablation studies with different configurations of the generators and discriminators in our proposed MPGAN. 1) We tested our model with locally-connected message passing, calculating the k-nearest neighbors in latent space before each layer of message passing. We varied the number of nearest neighbors and found lowering this significantly reduced both training stability and performance. 2) We tested the model without the masking strategy (using zero-padded and Gaussian-noise-padded particles) as well as several other masking strategies discussed in Appendix C. All resulted in lower performance than the chosen masking strategy described in Section 4. 3) As shown in Section 5, we tested our generator and discriminator separately in combination with previously studied discriminators and generators, respectively, and found the combination we propose outperforms the rest. All three studies support our final design choices, and the first two will be described and expanded upon in the supplementary materials along with quantitative scores for each metric.
>
>
> 3. Dealing with outliers
>
> Indeed, validating using aggregate metrics may not guarantee the lack of unrealistic outliers. This is in fact an issue with traditional Monte Carlo (MC) simulations as well, hence in every high energy physics analysis we perform rigorous checks, including comparisons in “control regions” (selected portions of the data with a known composition) to check for MC discrepancies with real data. Mismatches are corrected via reweighting the events (i.e. unrealistic simulated jets will be given less weight in the overall analysis) and are factored into the final uncertainties in the analysis results. The same procedure should be followed for data created by generative ML methods. Assuming the fraction of outliers is small (which is likely given the matching of the aggregate metrics), this reweighting will not have a significant effect on the uncertainty. We will incorporate this clarification about the downstream validation procedure in Section 3.1.

---

### Official Review · Reviewer_NRE3 · 2021-07-17

**Rating:** 6
**Confidence:** 3

**Summary:**

This paper introduces a new particle cloud dataset (JetNet) for high-energy physics. The authors set up a benchmark for particle cloud generation with a few physics-inspired and vision-related metrics. As existing point cloud GANs are not suitable for physics applications, they have also developed a new message-passing GAN (MPGAN) and have achieved good empirical performance.

**Limitations And Societal Impact:**

This paper does not have noticeable negative societal impacts. Besides, the authors have briefly discussed their limitations in Section 5.

**Main Review:**

The paper is well-written and easy to follow. The authors have provided sufficient physics background to help non-physics researchers better understand the problem setup. The topic studied in this paper (using machine learning to advance and accelerate physics research) is very interesting and has a large scientific impact. The authors have promised to release their dataset, which might benefit the researchers in the community a lot.

Apart from these mentioned in the related work (L117-L132), there have been a few other papers related to point cloud generation in the literature [Li et al., ICLR 2019 Workshop; Shu et al., ICCV 2019]. The authors have claimed that existing GANs are inadequate for physics applications. However, without sufficient comparisons with these existing methods, this argument is not very well justified. Besides, the authors have also mentioned in the introduction (L25-L31) that ML-based generative models might accelerate the simulation by five orders of magnitude. However, they have not provided any latency measurement on the hardware to support this claim.

This paper provides a bunch of different metrics for the particle cloud generation. It would be great if the authors could analyze the correlation between different metrics and provide some insights on which metric is the best, which might facilitate the comparisons between different methods in this direction. Besides, it is necessary to provide some qualitative examples to illustrate how the generated samples look like and some discussions on what the target downstream task is. It would be interesting to see how the generated point cloud performs in real physics applications.

**Post-Rebuttal Comments:**
The authors' rebuttal has addressed most of my concerns. I especially appreciate the additional results of TreeGAN and PC-GAN. Therefore, I will raise my score by 1 and vote for accepting this paper.

References:
1. Li et al., "Point Cloud GAN", ICLR 2019 Workshop.
2. Shu et al., "3D Point Cloud Generative Adversarial Network Based on Tree Structured Graph Convolutions", ICCV 2019.

**Time Spent Reviewing:**

3

---

> ### Author Response · Authors · 2021-08-11
> **Response to Reviewer 1**
>
> Thank you for the positive comments regarding the scientific impact of our paper, and the constructive suggestions. We address them and explain how the manuscript will be revised to incorporate them below.
>
> **Comparison with existing GAN methods**
>
> Thank you for this suggestion. We have now evaluated TreeGAN [1], both with a fully-connected and a PointNet-Mix discriminator, respectively, as well as PC-GAN [2] on our JetNet dataset. The preliminary results are given in the below table and a revised figure 4 for the paper featuring these models is linked (anonymously uploaded to Google Drive) below [Figure 1]. TreeGAN's upsampling and ancestor-descendant message passing technique is found not to be well-suited to our dataset, especially for variable-sized top and quark jets. PC-GAN uses a hierarchical-sampling-based architecture, assuming the conditional distributions of sampled points given a latent object representation are identical and independent. This assumption of independence does not hold for particle clouds where particles within a jet are highly correlated, and thus as expected PC-GAN performs poorly on this task. Hence these experiments support our claim that existing GANs are inadequate for physics applications, and will be included in the Related Work and Experiments sections in the final paper.
>
> | Class        | Gen     | Disc     | W1-P ($10^{-3}$) | W1-M ($10^{-3}$) | W1-EFP ($10^{-5}$) | FPND | COV  | MMD   |
> |--------------|---------|----------|--------------|--------------|----------------|------|------|-------|
> | Gluon        | TreeGAN | FC       | 13.1 ± 0.1   | 5.2 ± 0.2    | 4 ± 2          | 528  | 0.29 | 0.056 |
> |              | TreeGAN | PointNet | 2.8 ± 0.0    | 1.6 ± 0.2    | 4 ± 2          | 336  | 0.36 | 0.043 |
> |              | PCGAN   | PCGAN    | 29.5 ± 0.1   | 42.2 ± 0.2   | 14 ± 7         | 1.1k | 0.02 | 0.214 |
> | Top          | TreeGAN | FC       | 15.9 ± 0.1   | 17.0 ± 0.4   | 33 ± 16        | 176  | 0.38 | 0.084 |
> |              | TreeGAN | PointNet | 7.1 ± 0.1    | 5.0 ± 0.2    | 16 ± 7         | 258  | 0.52 | 0.078 |
> |              | PCGAN   | PCGAN    | 56.6 ± 0.2   | 111.2 ± 0.1  | 85 ± 40        | 625  | 0.02 | 0.348 |
> | Light  quark | TreeGAN | FC       | 14.4 ± 0.1   | 8.3 ± 0.3    | 5 ± 2          | 24k  | 0.14 | 0.044 |
> |              | TreeGAN | PointNet | 2.4 ± 0.1    | 4.5 ± 0.1    | 3 ± 1          | 23k  | 0.32 | 0.027 |
> |              | PCGAN   | PCGAN    | 45.3 ± 0.1   | 46.9 ± 0.1   | 9 ± 5          | 25k  | 0.01 | 0.920 |
>
> Figure 1: https://drive.google.com/file/d/1YB45HyuuwjKSTx1xGlT4bEn9tBb6eaWw/view?usp=sharing
>
> [1] Shu et al., 3D Point Cloud Generative Adversarial Network Based on Tree Structured Graph Convolutions, ICCV 2019.
>
> [2] Li et al., Point Cloud GAN, ICLR 2019 Workshop.
>
> &nbsp;&nbsp;
>
> **Hardware latency measurements**
>
> For this simplified dataset, the dominant CPU time is from the physics event generator (MadGraph) and hadron shower simulation (Pythia). We measure these on an 8-CPU machine as requiring approximately 46 ms per gluon jet. After this initial step, for JetNet we apply a simplified detector simulation and event reconstruction step, which requires negligible time. These steps are simplified in order to remain experiment-independent and allow public access to the dataset. However, in other more realistic scenarios, specific detector simulation and event reconstruction steps can add considerable time. For instance, simulation of the Large Hadron Collider’s CMS detector requires 12.3 s per top quark jet [3] and reconstruction requires roughly another 4 s per top quark jet [4]. In contrast, we measure the MPGAN approach to require 35.7 µs per jet, using an NVIDIA A100, which is already a significant three-orders-of-magnitude speed up on this dataset, and further could be used to replace the more time-intensive simulation and reconstruction steps as well, thus potentially rising to the quoted five orders of magnitude. We will add these experimental results in the Experiments section and expand Section 2 and Appendix A to include this clarification.
>
> [3] K. Pedro et al., EPJ Web Conf. 214, 02036 (2019), doi:10.1051/epjconf/201921402036
>
> [4] C. Chen et al., arXiv:2010.01835
>
> &nbsp;&nbsp;
>
> **Discussion of metrics**
>
> We note that we discuss the merits of each metric and provide suggestions for how to use them in the **Particle cloud evaluation metrics** paragraphs in Section 5, however, we recognize that we can further elaborate on this and the correlations between metrics. Therefore, we will add in Appendix E Figure 2, linked below (anonymously uploaded), showing correlation plots between metrics evaluated on 400 different MPGAN generated samples of 50,000 top quark jets each. Additionally we will incorporate into Section 5 and Appendix E the following discussion, building off of Section 5, on the correlation between individual metrics and clarifying our suggestions of the value and use cases of each.
>
> “We find, from Figure 2, W1-M and W1-EFP to be highly correlated, as expected since they both measure learning of global jet features. For rigorous validation we suggest measuring both but for time-sensitive use-cases, such as quick evaluations during model training, W1-M should be sufficient. W1-M, FPND, and W1-P are all measuring different aspects of the generation and are relatively uncorrelated. As discussed in Sec. 5, we expect FPND overall to be the best and most discriminatory metric for evaluation, while the W1 scores are valuable for their interpretability. Out of the latter, W1-M/W1-EFP are the most important from a physics-standpoint, as we generally characterize collisions by the high-level features of the output jets, rather than the individual particle features. MMD and COV are both valuable for specifically evaluating the quality and diversity of samples respectively, however we see from Fig. 2 that they saturate after a certain point, after which it is necessary to use the FPND and W1 scores. In conclusion, for thorough validation of generated particle clouds, we recommend considering FPND in conjunction with all three W-1 scores.”
>
> Figure 2: https://drive.google.com/file/d/1iiMGqvCG-Hdsw5kGvYlDQdEeR84LCmFe/view?usp=sharing
>
> &nbsp;&nbsp;
>
> **Qualitative examples of generated jets**
>
> Figures 3-5, below, showing sample real and generated gluon, quark, and top jets in discretized image-based representations in the angular-coordinate-plane, as well as an average jet image over 10,000 samples, will be added in Appendix D. We note that it is in general not easy to visually evaluate the quality of individual particle clouds (in contrast to e.g. ShapeNet point clouds), hence we focus in the paper on metrics aggregated over batches of clouds. We will clarify this point in Section 3.1.
>
> Figures 3-5: https://drive.google.com/drive/folders/1zmP_mFLK44WMER5anl-R7bGQCHHaWTUA?usp=sharing
>
> &nbsp;&nbsp;
>
> **Downstream physics task**
>
> Monte Carlo simulations, and potentially those produced by an ML generator, are used in experimental high energy physics to develop likelihood models for new physics theories potentially with unknown values of parameters of interest. We then compare these models with experimental data to perform hypothesis tests of the theories and estimate physical parameters (along with confidence intervals). Increasing the number of simulations and their accuracy can reduce the statistical uncertainties in our models thus allowing for higher precision measurements. The applications described here (physics model parameter estimation and hypothesis testing) are the main use cases that you inquire about. We will expand our introduction to further clarify this.

---

### Author Response · Authors · 2021-08-11
**General Response**

We thank all the reviewers for their thoughtful comments. We appreciate the overall positive feedback that the paper is “well-written” (R1),  “structured in a way that nicely introduces the problem setting” (R3), and that our application is “novel and interesting” (R2) and “has a large scientific impact” (R1), as well as acknowledging that “the discussion of design choices and merits/limitations of the metrics as well as the evaluation of all the models … is thorough” (R3). Regarding our proposed dataset, we are pleased to hear that ‘the introduction of JetNet is well-motivated and tackles a challenging problem in high energy physics’ (R3) and that the publishing of our dataset may “benefit researchers a lot” (R1) and “encourage researchers to contribute to the task” (R2).

The reviewers provided constructive suggestions to enhance the quality of the final manuscript which we address individually below. In summary:
  - We perform experiments with additional existing point cloud GANs (TreeGAN and PC-GAN) and find them to be inadequate for physics applications, and that our proposed MPGAN model is a significant improvement. Results shall be incorporated in Figure 4 and Table 2 of the paper.
  - We perform latency measurements to support our claim of order of magnitude improvements with machine learning over current Monte Carlo simulations, which will be added to the Experiments section.
  - We further discuss the merits of each of our proposed metrics and the correlations between them, supported by new correlation plots to be added to the appendix.
  - We will considerably strengthen the related works section by discussing the aforementioned two GANs as well as a variety of existing probabilistic models and their current limitations in the context of our JetNet dataset, noting that by publishing JetNet we hope to motivate and facilitate the further development of these models.
  - We will elaborate on the downstream physics application of the generated data in the introduction and evaluation metrics sections.
  - We will clarify the novel aspects of our model (physics-informed inductive biases, variable-sized cloud generation) and describe in more detail ablation studies to evaluate the effectiveness of individual design choices in the Architecture section and appendix.

---

### Decision · Program_Chairs · 2021-09-27

**Decision:**

Accept (Poster)

**Comment:**

All of the reviewers really liked this paper. They especially noted the clarity of exposition and the fact that the authors intend on open sourcing their data. During the rebuttal phase, the authors added new experimental results using TreeGAN that helped to address some concerns of the reviewers. Overall this seems like a timely and interesting contribution to the literature on point clouds and GANs.